# Misophonia impact questionnaire (MIQ), tinnitus impact questionnaire (TIQ), and hyperacusis impact questionnaire (HIQ): Factor analysis, test-retest reliability, and minimum detectable change using a non-clinical population

**Hashir Aazh**[1]*, **Fatma Betul Kula**[1,2]*

**1** Hashir International Specialist Clinics & Research Institute for Misophonia, Tinnitus and Hyperacusis, London, United Kingdom, **2** Department of Psychology, University of Surrey, Guildford, United Kingdom

* f.kula@surrey.ac.uk (FBK); info@hashirtinnitusclinic.com (HA)

## Abstract

The objectives of this study were to (1) assess if previously reported unidimensional factor structure of the Misophonia Impact Questionnaire (MIQ), Tinnitus Impact Questionnaire (TIQ), and Hyperacusis Impact Questionnaire (HIQ) can be confirmed in a nonclinical population, and (2) examine test-retest reliability and establish the minimum detectable change (MDC) for MIQ, TIQ and HIQ. 451 people completed the MIQ and HIQ sections of an online survey. A sub-sample of 173/451 who had tinnitus completed the TIQ too. 130/451 participants completed the second survey with 2 weeks interval for the MIQ and HIQ and 32/173 participants with tinnitus completed the TIQ in the second survey. All questionnaires showed excellent internal consistency, with Cronbach's α of 0.93 for both the MIQ and TIQ and 0.91 for the HIQ. Confirmatory factor analysis (CFA) showed that the MIQ, TIQ and HIQ were all one-factor questionnaires. Based on the intra-class correlation coefficients (ICC) values, the test-retest reliability was good for MIQ and HIQ and it was excellent for TIQ. Based on the MDC values, when these questionnaires are used for repeated measurements, the minimum amount of change that constitutes a true change is ≥ 8 for the total score of MIQ, ≥ 4 for the TIQ, and ≥ 7 for the total score of HIQ. In conclusion, the MIQ, TIQ and HIQ can be used in clinical practice or research setting to measure the impact of misophonia, tinnitus and hyperacusis on the individual's life, respectively, as one-factor questionnaires with excellent internal consistency and good to excellent test-retest reliability.

## Introduction

Although the impact of medical conditions on the sufferer's life is related to the severity of their symptoms, they are different constructs [1]. For example, several studies report

**Data availability statement:** The data that support the findings of this study are openly available in Zenodo at https://doi.org/10.5281/zenodo.15047909.

**Funding:** The author(s) received no specific funding for this work.

**Competing interests:** The authors have declared that no competing interests exist.

that the loudness of tinnitus is not the main factor contributing to how much tinnitus impacts on the patient's life [2,3]. Factors such as anxiety, depression, tinnitus annoyance are more strongly associated with tinnitus impact [4–6]. Misophonia is defined as reduced tolerance to certain sounds (known as the trigger sounds) such as the sounds associated with oral functions, nasal sounds, non-oral/nasal sounds produced by people, and sounds produced by objects or sounds generated by animals [7,8]. Several imaging studies shown that the individuals with misophonia have abnormal activation and functional connectivity of certain areas of their brain (namely the anterior insular cortex) when exposed to the trigger sounds [9,10]. Although the role of brain imaging studies in assessing misophonia has been challenged by some researchers [11], the observed abnormal brain activities among individuals with misophonia are consistent with their subjective ratings of the level of annoyance, anger and disgust caused by the trigger sounds as well as objectively recorded physiological responses (e.g., increased heart rate) [9,10]. However, the impact of misophonia on the life of the sufferers who might exhibit similar severity of brain activation/ physiological responses in the laboratory testing is likely to be different depending on their life style, coping methods and support available to them [12–15]. Hyperacusis is the perception of certain everyday sounds, such as domestic noise or noise in public places, as too loud or painful in such a way that it causes significant distress and impairment in social, occupational, recreational, and other day-to-day activities [16–18]. Hyperacusis has also shown to be related to increased sound-evoked responses in certain areas of the brain (namely supplementary motor area and cortical and subcortical auditory structures) [19,20]. Severity of hyperacusis can be estimated with the use of uncomfortable loudness levels (ULLs) test. People with hyperacusis often exhibit ULLs below 77 dB HL compared to the ULLs of around 100 dB HL for people with no hyperacusis [21,22]. However, studies shown that ULLs are weakly associated with the impact of hyperacusis on the patient's life [23–25]. To sum up, the construct of "impact on life" is different from the severity of symptoms of misophonia, tinnitus and hyperacusis.

In the absence of a definitive cure for misophonia, tinnitus and hyperacusis, various forms of cognitive behavioural therapies (CBT) [26–34], tinnitus and sound intolerance retraining therapy [35–38] and audiological counselling combined with sound therapy [39–41] are offered to patients by audiology and mental health professionals. The main aim of these interventions is to reduce the impact of the condition(s) on the patients' lives as opposed to reducing the severity of tinnitus loudness or diminishing any underlying physiological processes that can give rise to misophonia or hyperacusis. Therefore, in order to assess the efficacy of these interventions it is important to have questionnaires that focus on assessing the impact of misophonia, tinnitus and hyperacusis on the patient's life. The Misophonia Impact Questionnaire (MIQ) [42], Tinnitus Impact Questionnaire (TIQ) [43], and Hyperacusis Impact Questionnaire (HIQ) [44] are brief self-report questionnaires designed to assist clinicians and researchers to quantify the impact of misophonia, tinnitus and hyperacusis on the patient's life. The unique characteristic of the MIQ, TIQ, and HIQ is their focus on the construct of "impact on life" as opposed to assessing the severity of symptoms of misophonia, tinnitus and hyperacusis.

We previously reported psychometric properties of the MIQ [42], TIQ [43], and HIQ [44] using the data from patients seeking help for distressing misophonia, tinnitus and hyperacusis from an specialist outpatient audiology clinic in the UK. The confirmatory factor analysis (CFA) in the previous studies revealed that a one-factor model for the MIQ and HIQ to assess the constructs of misophonia impact and hyperacusis impact, respectively, gave excellent fits [42,44]. The CFA for the TIQ suggested a bi-factor model with sufficient unidimensionality to support the use of the overall TIQ score for assessing the impact of tinnitus [43]. However, these studies were based on the samples taken from patients seeking help from a specialist audiology outpatient clinic. The problem with using clinical populations is that the majority of patients seeking medical help for their misophonia, tinnitus, and hyperacusis often experience significant impact of their conditions on their lives, leading to a small variety of scores for the questionnaires' items (i.e., patients giving higher or maximum scores indicating significant impact to the most of the questionnaires' items). In contrast, samples taken from non-clinical populations (i.e., people who either do not have a diagnosed misophonia/tinnitus/hyperacusis or are not currently seeking help or receiving treatment for them) vary in the degree of impact of the condition on their lives. This provides information about how the questionnaires perform among patients with more diverse degrees of misophonia, tinnitus and hyperacusis impacts [45]. The first aim of the present study is to explore if the unidimensionality of the MIQ, TIQ, and HIQ can be confirmed in a non-clinical population.

Our second aim is to assess the test-retest reliability of MIQ, TIQ, and HIQ which estimates the consistency of the questionnaires when used for repeated measurements by calculating the intraclass correlation coefficients (ICC). Using a sample from a non-clinical population is desirable for this purpose as the vast majority of participants are not expected to be receiving active treatment between the test and retest measurements.

As the MIQ, TIQ, and HIQ were designed to be used for assessing the efficacy of treatments such as CBT, it is important to establish their minimal detectable change (MDC). Currently, there is no amount of change in the scores of MIQ, TIQ and HIQ that can be considered as a reliable change and the decision is often based on the subjective interpretation of clinicians or researchers. The MDC can be used to identify the minimal amount of change in the scores of a questionnaire that distinguishes a true change from a change due to the measurement error [46]. The MDC assists clinicians and researchers to better interpret the change in the questionnaire scores before and after a particular intervention. Our third aim is to assess MDC for MIQ, TIQ and HIQ.

To sum up, the aims of this study are to (1) assess if previously reported unidimensional factor structures of MIQ, TIQ, HIQ can be confirmed in a nonclinical population, (2) examine test-retest reliability of the questionnaires, and (3) establish minimum detectable change for MIQ, TIQ and HIQ.

## Methods

### Ethical approval

The ethical approval was granted by the University of Surrey Research Integrity and Governance Office (Project ID: FHMS 21–22 083). Consent was obtained electronically through the online survey platform, where participants were presented with an information sheet detailing the study's purpose, procedures, and their rights. Participants confirmed their consent by selecting a checkbox indicating their agreement to participate.

### Study design and participants

This was a cross-sectional study using an online survey in which the factor analysis, test-retest reliability and MDC of the self-administered MIQ, TIQ and HIQ were examined using Qualtrics platform (Qualtrics, Provo, USA, https://www.qualtrics.com). The survey was conducted between 1 May and 28 July 2023, with participants invited to complete it twice at a two-week interval. The inclusion criteria comprised: age ≥ 18 years old and good understanding of English language. Participants were excluded if they had a significant cognitive impairment or severe visual impairment (this was communicated

with them in the study invitation and information sheet). 451 people completed the MIQ and HIQ sections of the first online survey. A sub-sample of 173/451 who had tinnitus completed the TIQ section of the first survey in addition to the MIQ and HIQ. Although all participants were sent the link to the second survey, only 130 participants (out of 451 who completed the MIQ and HIQ in the first survey) completed the second survey with 2 weeks interval for the MIQ and HIQ and 32 participants (out of 173 who had tinnitus and completed TIQ in the first survey) completed the TIQ in the second survey. 69.6% (314/451) were recruited from students and staff of University of Surrey, and the remaining 137 (30.4%) were recruited from emailing to misophonia, tinnitus and hyperacusis charity groups and social media. The age of the participants ranged between 18–86 years old with mean and standard deviation (SD) of 36.5 years (12.8 years). 60.3% (N = 272) were female, 37.5% (N = 169) were male, and 2.2% were self-identified as non-binary (N = 10). The highest level of education was bachelor's degree or post graduate qualifications in 69% of the participants. Majority of the participants (76.7%) identified their ethnicity as white British, Irish, and other white ethnic backgrounds. With regard to the participants' employment status, 72% were employed, 18.4% were students, 4.2% retired, 3.1% unemployed, and 2.2% other (self-employed, disability, stay-at-home mom, sick, unable to work).

## Questionnaires

**Misophonia impact questionnaire (MIQ).** MIQ is an eight-item questionnaire. It assesses the impact of misophonia on life by asking the patient to indicate, over the last 2 weeks, how often they have experienced the problems described in *items* 1–8 because of their intolerance to certain sounds (i.e., sounds related to eating, chewing gum, lip smacking, mouth noises, sniffling, breathing, clicking, and tapping). The items are: (1) Feeling anxious, (2) Unable to distract yourself from certain sounds, (3) Experiencing difficulties in your relationships with family members or friends, (4) Feeling angry, (5) Finding it difficult to be around certain individuals because of the noises that they make, (6) Feeling irritated, (7) Avoiding certain situations because of the noises that you have to put up with, and (8) Experiencing low mood because of your intolerance to certain sounds. For each item, a score of 0, 1, 2, or 3 is assigned to the response categories of "0 to 1 days", "2 to 6 days", "7 to 10 days", and "11 to 14 days", respectively. The total score of MIQ is calculated by the sum of the scores for the eight items and it ranges between 0 and 24. Its Cronbach's α is 0.94 [42]. See the MIQ in appendix 1.

**Tinnitus impact questionnaire (TIQ).** TIQ is a seven-item questionnaire. It assesses the impact of tinnitus on life by asking the patient to indicate, over the last 2 weeks, how often they have experienced the problems described in items 1–7 because of their tinnitus. The items are: (1) Lack of concentration, (2) Feeling anxious, (3) Sleep difficulties (delay in falling asleep and/or difficulty getting back to sleep if woken up during the night), (4) Lack of enjoyment from leisure activities, (5) Inability to perform certain day-to-day activities/tasks, (6) Feeling irritable, and (7) Low mood. For each item, a score of 0, 1, 2, or 3 is assigned to the response categories of "0 to 1 days", "2 to 6 days", "7 to 10 days", and "11 to 14 days", respectively. The response choices for item 3 that assesses sleep difficulties are "0 to 1 nights", "2 to 6 nights", "7 to 10 nights", and "11 to 14 nights", The total score of TIQ is calculated by the sum of the scores for the seven items and it ranges between 0 and 21. Its Cronbach's α is 0.84 [43]. See the TIQ in appendix 2.

**Hyperacusis impact questionnaire (HIQ).** HIQ is an eight-item questionnaire. It assesses the impact of hyperacusis on life by asking the patient to indicate, over the last 2 weeks, how often they have experienced the problems described in items 1–8 because of certain environmental sounds which seem too loud to them but other people could tolerate them well. The items are (1) Feeling anxious when hearing loud noises, (2) Avoiding certain places because it is too noisy, (3) Lack of concentration in noisy places, (4) Unable to relax in noisy places, (5) Difficulty in carrying out certain day-to-day activities/tasks in noisy places, (6) Lack of enjoyment from leisure activities in noisy places, (7) Experiencing low mood because of your intolerance to sound, and (8) Getting tired quickly in noisy places. For each item, a score of 0, 1, 2, or 3 is assigned to the response categories of "0 to 1 days", "2 to 6 days", "7 to 10 days", and "11 to 14 days", respectively. The total score of HIQ is calculated by the sum of the scores for the eight items and it ranges between 0 and 24. Its Cronbach's α is 0.93 [44]. See the HIQ in appendix 3.

## Statistical analysis

The structural validity of the MIQ, TIQ, and HIQ were tested by confirmatory factor analysis (CFA). The values required for good model fit comprise: Root Mean Square Error of Approximation (RMSEA) ≤ 0.05, Standardized Root Mean Square Residual (SRMR) ≤ 0.05, Tucker Lewis Index (TLI) ≥ 0.97, Comparative Fit Index (CFI)≥0.97, and CMIN (Chi-square/ df) of close to 2 [47]. CFA was conducted using the lavaan package in R (version 0.6–24) to evaluate the latent structure of the scales. Given that item responses were provided on a scale with ordered categories, the weighted least squares mean variance adjusted (WLSMV) estimator was employed [48]. For the structures to be considered satisfactory, factor loadings were required to be at least 0.50 for each item [48].

Test-retest reliability was assessed using the intra-class correlation coefficients (ICC) calculated for the total scores of the questionnaires between the first and the second surveys with a two-week interval. The ICC is a value between 0 and 1, where values below 0.5 indicate poor reliability, between 0.5 and 0.75 moderate reliability, between 0.75 and 0.90 good reliability, and values above 0.90 indicate excellent reliability [49]. T-tests were used for comparing means between test and retest scores. The MDC is calculated using the standard error of measurement (SEM) based on the SD of the difference in the total scores of MIQ, TIQ and HIQ between the test and retest surveys. The SEM and MDC were calculated with the following formulae:

$$\text{SEM} = \text{SD}/\sqrt{2} \text{ and MDC (with 95\% confidence interval)} = \text{SEM} \times \sqrt{2} \times 1.96$$

Only the data for participants who completed for both surveys were used for calculating ICC and MDC.

Internal consistency was measured with Cronbach's $\alpha$ and McDonald's $\omega$ [50] for which values greater than 0.7 is considered acceptable [51]. The value of $\alpha$ when each item was deleted (AID) and the item-total correlations (ITC) were calculated. The ITC for a given item is the correlation between scores for that item and scores for the total excluding that item. Values between 0.3 and 0.8 are required [52] for acceptable ITC. Pearson's correlation coefficients were calculated to assess the relationships between age and total scores of the TIQ, MIQ, and HIQ.

Statistical analyses, apart from CFA were conducted using IBM SPSS software, and Amos version 28.0. The number of participants (N) included to each analysis is reported when indicated.

## Results

### Confirmatory factor analysis

One factor model for MIQ, TIQ, and HIQ gave a good fit for most measures of goodness-of-fit as shown in Table 1. The RMSEA values met the criteria of ≤ 0.05 for MIQ and TIQ and it was 0.06 for HIQ which is slightly above the required value. The SRMR values met the criteria of ≤ 0.05 for all questionnaires. The CFI and TLI were above 0.95 for all questionnaires. The CMIN was close to 2 for all questionnaires. The standardised factor loadings ranged from 0.81 to 0.89 for MIQ, from 0.70 to 0.88 for TIQ, and from 0.67 to 0.76 for the HIQ indicating no problematic items and a suitable solution (Table 2). Based on the value of $\alpha$ if item was deleted (AID), deleting any of the items did not improve $\alpha$ hence it was decided that all of the items should be retained in the MIQ, TIQ and HIQ scales. Overall, these findings support a good fit for the one factor model for the MIQ, TIQ and HIQ.

### Reliability

Both Cronbach's α and McDonald's ω were 0.93 for the MIQ and TIQ and they were 0.91 for the HIQ which indicate high level of internal consistency. Values of ITC for items of the 3 questionnaires were between 0.64 and 0.83 which are acceptable (Table 2). As shown in Table 3, there were no statistically significant differences in the mean scores for the MIQ, TIQ, and HIQ between the first and the second surveys which were taken with a two-weeks interval. Based on the ICC values, the test-retest reliability was rated as good for the total score of MIQ and HIQ and it was rated as excellent reliability for the TIQ.

**Table 1. Goodness-of-fit indices for one factor models for MIQ, TIQ, and HIQ.**

| Fit Statistics | MIQ (N = 451) | TIQ (N = 173) | HIQ (N = 451) |
|---|---|---|---|
| RMSEA | 0.04 | 0.05 | 0.06 |
| SRMR | 0.01 | 0.04 | 0.03 |
| CFI | 0.99 | 0.99 | 0.98 |
| TLI | 0.99 | 0.98 | 0.97 |
| CMIN (x²/df) | 2.02 | 1.41 | 2.61 |

Note. MIQ: Misophonia Impact Questionnaire; TIQ: Tinnitus Impact Questionnaire; HIQ: Hyperacusis Impact Questionnaire; CMIN: Chi-square/ df; RMSEA: root mean squared error of approximation; CFI: comparative fit index; TLI: Tucker-Lewis index; SRMR: standardized root mean squared residual; N: number of participants.

**Table 2. Standardised factor loadings and reliability statistics for the MIQ, TIQ and HIQ.**

| Questionnaire | Item | Loading | ITC | AID |
|---|---|---|---|---|
| MIQ | MIQ1 | 0.85 | 0.76 | 0.92 |
| | MIQ2 | 0.87 | 0.78 | 0.92 |
| | MIQ3 | 0.81 | 0.72 | 0.93 |
| | MIQ4 | 0.84 | 0.75 | 0.92 |
| | MIQ5 | 0.89 | 0.81 | 0.92 |
| | MIQ6 | 0.87 | 0.78 | 0.92 |
| | MIQ7 | 0.81 | 0.72 | 0.93 |
| | MIQ8 | 0.86 | 0.77 | 0.92 |
| TIQ | TIQ1 | 0.75 | 0.72 | 0.92 |
| | TIQ2 | 0.85 | 0.81 | 0.92 |
| | TIQ3 | 0.70 | 0.67 | 0.92 |
| | TIQ4 | 0.83 | 0.79 | 0.92 |
| | TIQ5 | 0.78 | 0.74 | 0.92 |
| | TIQ6 | 0.88 | 0.83 | 0.92 |
| | TIQ7 | 0.85 | 0.80 | 0.93 |
| HIQ | HIQ1 | 0.67 | 0.64 | 0.91 |
| | HIQ2 | 0.69 | 0.66 | 0.91 |
| | HIQ3 | 0.76 | 0.72 | 0.91 |
| | HIQ4 | 0.76 | 0.72 | 0.91 |
| | HIQ5 | 0.76 | 0.72 | 0.91 |
| | HIQ6 | 0.76 | 0.72 | 0.91 |
| | HIQ7 | 0.74 | 0.71 | 0.91 |
| | HIQ8 | 0.75 | 0.71 | 0.91 |

Note. MIQ: Misophonia Impact Questionnaire; TIQ: Tinnitus Impact Questionnaire; HIQ: Hyperacusis Impact Questionnaire; ITC: item-total correlation; AID: α if item deleted.

Based on the MDC values, when these questionnaires are used for repeated measurements, the minimum amount of change that constitutes a true change is ≥ 8 for the total score of MIQ, ≥ 4 for the TIQ, and ≥ 7 for the total score of HIQ (Table 3). The Pearson correlation analysis revealed that there were no significant correlations between age and the total scores of the questionnaires: HIQ (r = −0.60, p > 0.05), MIQ (r = −0.08, p > 0.05), and TIQ (r = −0.10, p > 0.05).

**Table 3. The table shows the means (M) and standard deviations (SD) for the total scores of MIQ, TIQ, HIQ, and SAD-T calculated for the first and the second times that participants completed the survey for assessment of test-retest reliability (N = 154). In addition, the table shows the Interclass Correlation Coefficient (ICC) values and their 95% Confidence Intervals (CI), the Standard Error of Measurement (SEM) and the Minimum Detectable Change (MDC) for each questionnaire.**

| | First test M (SD) | Retest M (SD) | t p 95% CI | ICC [95% CI] | SEM | MDC |
|---|---|---|---|---|---|---|
| MIQ (N = 130) | 9.69 (6.38) | 9.70 (6.37) | −0.01 0.98 [-0.92, 0.91] | 0.79 [0.71, 0.86] | 2.92 | 8.09 |
| TIQ (N = 32) | 6.36 (6.42) | 6.79 (4.47) | −0.73 0.47 [−1.63, 0.77] | 0.94 [0.87, 0.97] | 1.36 | 3.77 |
| HIQ (N = 130) | 9.88 (6.09) | 9.23 (6.29) | 1.64 0.10 [−0.13, 1.42] | 0.84 [0.78, 0.89] | 2.48 | 6.87 |

Note. MIQ: Misophonia Impact Questionnaire; TIQ: Tinnitus Impact Questionnaire; HIQ: Hyperacusis Impact Questionnaire; N: number of participants; p: p-value; t: t statistics with degree of freedom equal to 129, 27, 129 for MIQ, TIQ, and HIQ, respectively; N: number of participants.

## Discussion

This study was conducted using a non-clinical population compared to the previous research on these questionnaires which used the data from clinical populations (i.e., patients seeking help for their misophonia, tinnitus, or hyperacusis from a specialist centre) [42–44,53,54]. Majority of participants in the present survey (about 70%) were recruited from students and staff in one UK university. The number of participants who completed MIQ and HIQ were 451 and the number of participants who completed TIQ was 173 which are more than the minimum sample size of 150 required for factor analysis [55]. The standardised factor loadings in this study were 0.81–0.89 for the items of MIQ which are comparable to the factor loadings of 0.86–0.99 reported in a clinical population [42]. The factor loadings were 0.70–0.88 for the TIQ which are comparable to values of 0.51–0.87 and 0.67–0.89 reported in two different studies using clinical populations [43,54]. The factor loadings were 0.67–0.76 for the HIQ which is slightly lower than 0.77 to 0.96 reported in a clinical population [44]. Overall, the factor loadings for the items of these three questionnaires were above 0.5, indicating no problematic items and a suitable solution. The CFA showed that the MIQ, TIQ, and HIQ can be used as one-factor questionnaires to assess the impact of misophonia, tinnitus and hyperacusis, respectively, on the individuals' life. This is consistent with most of the studies conducted in clinical populations [42,44,54]. Therefore, the total score of the MIQ, TIQ and HIQ can be used in day-to-day clinical practice as well as in the surveys among the general population. Contrary to the results of the present study, Aazh *et al.* [43] reported that the one-factor solution for the TIQ gave an inadequate fit, with TIQ3 (sleep difficulties) and TIQ5 (inability to perform certain day-to-day activities/tasks) having low loadings indicating that these items did not contribute substantially to the overall measurement of the latent construct. They reported that a bi-factor model gave a better fit than the one-factor model, with a relative χ2 value of 2.64 and factor loadings of 0.51–0.87. In a bi-factor model, the total score of the questionnaire can still be interpreted as a one-dimensional measure of the construct in question as all of the items are sufficiently unidimensional, with some items have a degree of construct-relevant multidimensionality [56,57]. It is possible that as Aazh *et al.* [43] used data from a clinical population in an outpatient clinic, the majority of their patients experienced significant tinnitus-induced sleep problems and difficulty to perform certain day-to-day activities. Hence there were small range of scores for the TIQ3 and TIQ5, impacting on their correlations with other items and factor loadings. Use of a tinnitus sample from the general population in the present study might have led to a wider range of responses for items TIQ3 and TIQ5 and consequently higher item-total correlations and better factor loadings.

The total scores of MIQ, TIQ and HIQ have good to excellent test-retest reliability hence they can be used in clinical practice or research to assess the change in self-report impact of misophonia, tinnitus and hyperacusis, respectively, before and after a treatment. However, it is important to take into consideration the values of MDC which demonstrate the minimum change in the questionnaire score that is required to reflect a true change beyond the measurement error of the questionnaire when used for repeated measurements [46]. The MDC was about 8 for MIQ, 4 for TIQ and 7 for HIQ. This indicates that people's report of the impact of misophonia and hyperacusis (without any intervention) in a two weeks period seem to change more than that of tinnitus impact. This observation can also be related to a smaller sample size for individuals completed the TIQ twice compared to that of MIQ and HIQ, 32 vs 130. For assessment of test-retest reliability and the MDC, a minimum of 100 participants is recommended [58]. Therefore, the MDC and ICC reported for the TIQ should be interpreted with caution and future studies with larger sample are needed to further assess the performance of the TIQ during repeated measurements. Another limitation of this study is its cross-sectional design. While some of the participants have completed the questionnaires twice (130 for the MIQ and HIQ and 32 for the TIQ), longitudinal data from people who receive treatments for misophonia, tinnitus and hyperacusis are needed in order to establish the sensitivity of MIQ, TIQ, and HIQ to change. This will require large clinical trials using the MIQ, TIQ, and HIQ in parallel with other established outcome measures. Future studies should also explore minimally important change (MIC) which allows clinicians to interpret if changes in the scores of MIQ, TIQ and HIQ can be considered as an important clinical change [59]. There are child versions for the MIQ, TIQ, and HIQ which are designed to be completed by parents of children aged 16 years and younger (appendixes 4–6) [42,53]. Future studies should assess test-retest reliability and other psychometric properties for the child versions of these questionnaires.

## Conclusions

Assessing the impact of misophonia, tinnitus, and hyperacusis on the patients' lives and monitoring the effect of treatment on such impact are critical. Having brief self-administered measures to complete either in the clinic or online would save clinical time and resources. Brevity, robust factor structure, and excellent internal consistency make the MIQ, TIQ and HIQ attractive instruments for assessing and monitoring the impact of these conditions. Future studies should assess their sensitivity to change based longitudinal data from clinical trials using the MIQ, TIQ and HIQ as outcome measures and to explore minimally important change for these questionnaires.

## Supporting information

**S1 Appendix:  Misophonia Impact Questionnaire.**
(DOCX)

**S2 Appendix.  Tinnitus Impact Questionnaire.**
(DOCX)

**S3 Appendix.  Hyperacusis Impact Questionnaire.**
(DOCX)

**S4 Appendix.  Misophonia Impact Questionnaire (parent version).**
(DOCX)

**S5 Appendix.  Tinnitus Impact Questionnaire (parent version).**
(DOCX)

**S6 Appendix.  Hyperacusis Impact Questionnaire (Parent version).**
(DOCX)

## Acknowledgments

This project was based on the data collected for FBK's PhD research at University of Surrey.

## Author contributions

**Conceptualization:** Hashir Aazh, Fatma Betul Kula.

**Formal analysis:** Fatma Betul Kula.

**Methodology:** Fatma Betul Kula.

**Project administration:** Fatma Betul Kula.

**Supervision:** Hashir Aazh.

**Writing – original draft:** Hashir Aazh.

**Writing – review & editing:** Hashir Aazh.

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
