## [Decision Letter · Decision Letter 0]

7 Feb 2025

PONE-D-24-56831Misophonia Impact Questionnaire (MIQ), Tinnitus Impact Questionnaire (TIQ), and Hyperacusis Impact Questionnaire (HIQ): Factor Analysis, Test-Retest Reliability, and Minimum Detectable Change Using a Non-Clinical PopulationPLOS ONE

Dear Dr. Kula,

Thank you for submitting your manuscript to PLOS ONE. After careful consideration, we feel that it has merit but does not fully meet PLOS ONE’s publication criteria as it currently stands. Therefore, we invite you to submit a revised version of the manuscript that addresses the points raised during the review process.

We look forward to receiving your revised manuscript.

Kind regards,

Prashanth Prabhu

Academic Editor

PLOS ONE

Journal Requirements:

2. Thank you for stating the following in the Acknowledgments Section of your manuscript: [HA was supported by an R&D fund from Hashir International Specialist Clinics & Research Institute for Misophonia, Tinnitus and Hyperacusis Ltd. This project was based on the data collected for FBK’s PhD research at University of Surrey, funded by the Ministry of National Education of the Republic of Turkey.]

Please remove any funding-related text from the manuscript and let us know how you would like to update your Funding Statement. Currently, your Funding Statement reads as follows: “The authors received no specific funding for this work.”

3. We note that you have indicated that there are restrictions to data sharing for this study. For studies involving human research participant data or other sensitive data, we encourage authors to share de-identified or anonymized data. However, when data cannot be publicly shared for ethical reasons, we allow authors to make their data sets available upon request. For information on unacceptable data access restrictions, please see http://journals.plos.org/plosone/s/data-availability#loc-unacceptable-data-access-restrictions .

b) If there are no restrictions, please upload the minimal anonymized data set necessary to replicate your study findings to a stable, public repository and provide us with the relevant URLs, DOIs, or accession numbers. Please see http://www.bmj.com/content/340/bmj.c181.long for guidelines on how to de-identify and prepare clinical data for publication. For a list of recommended repositories, please see https://journals.plos.org/plosone/s/recommended-repositories . You also have the option of uploading the data as Supporting Information files, but we would recommend depositing data directly to a data repository if possible.

4. Please include captions for your Supporting Information files at the end of your manuscript, and update any in-text citations to match accordingly. Please see our Supporting Information guidelines for more information: http://journals.plos.org/plosone/s/supporting-information .

Reviewers' comments:

Reviewer's Responses to Questions

**Comments to the Author**

1. Is the manuscript technically sound, and do the data support the conclusions?

Reviewer #1: Yes

Reviewer #2: Yes

2. Has the statistical analysis been performed appropriately and rigorously? 

Reviewer #1: Yes

Reviewer #2: Yes

3. Have the authors made all data underlying the findings in their manuscript fully available?

Reviewer #1: Yes

Reviewer #2: Yes

4. Is the manuscript presented in an intelligible fashion and written in standard English?

Reviewer #1: Yes

Reviewer #2: Yes

5. Review Comments to the Author

Reviewer #1: Thank you for this comprehensive paper on the questionnaires used to evaluate the impact of Tinnitus, Misophonia, and Hyperacusis.

Minimal revision is advised.

1. Line 204, Page 8 :There is a typo: "completed for bother surveys were...". I assume the authors meant "both."

2. Could the authors please clarify the process of participant recruitment? From my understanding, 451 participants completed the two questionnaires, the MIQ and HIQ, and 173 of them completed the TIQ. Furthermore, 130 participants who repeated the questionnaire also completed both MIQ and HIQ, while 32 of the 173 participants completed the TIQ.

3. Did the 451 participants report symptoms of misophonia or hyperacusis to be included in the study? In other words, what were the requirements for participant inclusion?

Reviewer #2: PONE-D-24-56831: Misophonia Impact Questionnaire (MIQ), Tinnitus Impact Questionnaires (TIQ), and Hyperacusis Impact Questionnaire (HIQ): Factor analysis, Test-retest reliability, and minimum detectable change using a non-clinical population

This manuscript is a sequel of the author’s previous work on the development and validation of the three questionnaires. The authors assessed the factor analysis, test-retest reliability, and also identified the minimum detectable change for the 3 questionnaires when completed on a non-clinical population. This topic is relevant for the journal and will provide relevant information to better assess individuals with tinnitus or other loudness issues, especially to identify the impacts of those conditions in the patient’s life. However, this manuscript, in its current format, needs some restructuring, clarity, and cohesiveness for ease of understanding and replicability. I suggest the authors consider revising the manuscript if they wish to re-submit it. See below my comments.

Abstract:

It was later mentioned in the manuscript that a sub-set of the total sample completed the surveys twice to assess the test-retest reliability. It is not currently clear based on how it is written in lines 21-22. Consider editing the sentence to clarify this information

Introduction:

The introduction section has a lot of great, relevant information. However, the way it is currently structured assumes that the readers are already aware of the authors’ previous work. I suggest the authors provide a better orientation to this study and rearrange the section for better cohesiveness and ease of understanding.

For example:

The current paragraph 2 (lines 51-77) can be the first paragraph as it introduces the conditions and provides more information about them.

Lines 78 to 86: This can be paragraph 2 as it flows better after the introductory information. Additionally, consider beginning this paragraph with an introductory sentence to the paragraph so it flows better.

Lines 37-50: Once you discussed about interventions and their aims and the importance of assessment. This current first paragraph fits better as it talks about the questionnaires how they are in their existing original format. You can consider ending this paragraph with the sentence in line 86 (The MIQ….purpose.).

Lines 87 onwards: This can be the 4th paragraph of the introduction section as this then talks about the need to test the questionnaires in a non-clinical population. I would also suggest providing a straightforward description of the meaning of non-clinical population as you introduce this term in this paragraph. This will allow the readers to avoid any confusion in interpreting who you are referring to as this population.

Lin2 108: The sentence “Currently..” started talking about the 3rd aim. I wasn’t sure about that until read the last sentence of the paragraph. I suggest, editing this sentence slightly to indicate that you have started talking about the 3rd aim.

Methods:

Line 132-133: The authors have mentioned about the subset of population who completed the survey twice. Were these individuals selected and invited for the re-test survey or was the re-test survey was available to the 451 individuals and these 130 individuals voluntarily completed the survey twice. Consider clarifying this information.

Line 134: Add the % for the 137 individuals

Line 136: The authors have included a wide age range. Consider providing a rationale for this choice. Were there any age-related differences in the results?

Consider providing more details on all the items in the survey, other than the 3 questionnaires. For example, did they provide their demographic details?

Lines 142-145: This is a really long sentence. Consider chunking it into two sentences for ease of understanding.

Line 204-205: Only 32 participants’ data was included. Do the authors have information on the power of the study with this limited sample?

Results:

Lines 215-224: To improve ease of reading for the readers, consider following the same order for the statistics and results as they are presented in the table.

Discussion:

Consider discussing more on the clinical implication of your results in individuals with these conditions.

Conclusion:

Consider providing a stronger conclusion beyond a summary of the results.

6. PLOS authors have the option to publish the peer review history of their article (what does this mean? ). If published, this will include your full peer review and any attached files.

**Do you want your identity to be public for this peer review?** For information about this choice, including consent withdrawal, please see our Privacy Policy .

Reviewer #1: No

Reviewer #2: **Yes: ** Lipika Sarangi

---

## [Author Response · Author response to Decision Letter 0]

5 Apr 2025

Dear Dr Prashanth Prabhu

We describe below the changes that we have made in response to the comments of the editor and reviewers.

Best regards

Hashir Aazh and Fatma B. Kula

Journal Requirements:

Answer: We have followed all of PLOS ONE's style requirements and templates.

2. Please remove any funding-related text from the manuscript and let us know how you would like to update your Funding Statement. Currently, your Funding Statement reads as follows: “The authors received no specific funding for this work.”

Answer: We have removed any funding-related text from the Acknowledgements section as requested. We would like to leave the funding statement unchanged as “The authors received no specific funding for this work.”

Answer: The data has been published on Zenodo without ethical or legal restrictions, and the data availability statement has been updated accordingly.

4. Please include captions for your Supporting Information files at the end of your manuscript, and update any in-text citations to match accordingly.

Answer: We have already shared our data on the recommended open access data sharing platform from PLOS One, so we did not provide additional Supporting Information files.

Reviewers' comments:

Reviewer #1:

Minimal revision is advised.

1. Line 204, Page 8 :There is a typo: "completed for bother surveys were...". I assume the authors meant "both."

Answer: Done

2. Could the authors please clarify the process of participant recruitment? From my understanding, 451 participants completed the two questionnaires, the MIQ and HIQ, and 173 of them completed the TIQ. Furthermore, 130 participants who repeated the questionnaire also completed both MIQ and HIQ, while 32 of the 173 participants completed the TIQ.

Answer: We have now made this clearer with in the section of Study Design and Participants.

3. Did the 451 participants report symptoms of misophonia or hyperacusis to be included in the study? In other words, what were the requirements for participant inclusion?

Answer: No. The inclusion criteria were being an adult and speaking English. We have now explained this in the section of Study Design and Participants.

Reviewer #2:

Abstract:

It was later mentioned in the manuscript that a sub-set of the total sample completed the surveys twice to assess the test-retest reliability. It is not currently clear based on how it is written in lines 21-22. Consider editing the sentence to clarify this information

Answer: We have now clarified this in the abstract.

Introduction:

The introduction section has a lot of great, relevant information. However, the way it is currently structured assumes that the readers are already aware of the authors’ previous work. I suggest the authors provide a better orientation to this study and rearrange the section for better cohesiveness and ease of understanding.

For example:

The current paragraph 2 (lines 51-77) can be the first paragraph as it introduces the conditions and provides more information about them.

Answer: done

Lines 78 to 86: This can be paragraph 2 as it flows better after the introductory information. Additionally, consider beginning this paragraph with an introductory sentence to the paragraph so it flows better.

Answer: done

Lines 37-50: Once you discussed about interventions and their aims and the importance of assessment. This current first paragraph fits better as it talks about the questionnaires how they are in their existing original format. You can consider ending this paragraph with the sentence in line 86 (The MIQ….purpose.).

Answer: done

Lines 87 onwards: This can be the 4th paragraph of the introduction section as this then talks about the need to test the questionnaires in a non-clinical population. I would also suggest providing a straightforward description of the meaning of non-clinical population as you introduce this term in this paragraph. This will allow the readers to avoid any confusion in interpreting who you are referring to as this population.

Answer: done

Lin2 108: The sentence “Currently..” started talking about the 3rd aim. I wasn’t sure about that until read the last sentence of the paragraph. I suggest, editing this sentence slightly to indicate that you have started talking about the 3rd aim.

Answer: done

Methods:

Line 132-133: The authors have mentioned about the subset of population who completed the survey twice. Were these individuals selected and invited for the re-test survey or was the re-test survey was available to the 451 individuals and these 130 individuals voluntarily completed the survey twice. Consider clarifying this information.

Answer: We have now clarified this.

Line 134: Add the % for the 137 individuals

Answer: Done

Line 136: The authors have included a wide age range. Consider providing a rationale for this choice. Were there any age-related differences in the results?

Answer: We have added the inclusion criteria to the methods section which clarifies that anyone with age≥ 18 years old was eligible. These questionnaires were designed to be used by adults regardless of their age. We have now added the correlation between age and the scores on MIQ, TIQ and HIQ as suggested by the reviewer.

Consider providing more details on all the items in the survey, other than the 3 questionnaires. For example, did they provide their demographic details?

Answer: We have now added the stats about education, ethnicity, and employment status of the participants.

Lines 142-145: This is a really long sentence. Consider chunking it into two sentences for ease of understanding.

Answer: Done

Line 204-205: Only 32 participants’ data was included. Do the authors have information on the power of the study with this limited sample?

Answer: We have discussed this as a limitation with the discussion section:

“This observation can also be related to a smaller sample size for individuals completed the TIQ twice compared to that of MIQ and HIQ, 32 vs 130. For assessment of test-retest reliability and the MDC, a minimum of 100 participants is recommended [62]. Therefore, the MDC and ICC reported for the TIQ should be interpreted with caution and future studies with larger sample are needed to further assess the performance of the TIQ during repeated measurements.”

Results:

Lines 215-224: To improve ease of reading for the readers, consider following the same order for the statistics and results as they are presented in the table.

Answer: Done

Discussion:

Consider discussing more on the clinical implication of your results in individuals with these conditions.

Answer: We have added more discussions about clinical use and limitations of using these in clinical practice for assessment of treatment effect.

Conclusion:

Consider providing a stronger conclusion beyond a summary of the results.

Answer: We have now reworded the conclusions.

---

## [Decision Letter · Decision Letter 1]

30 Apr 2025

Misophonia Impact Questionnaire (MIQ), Tinnitus Impact Questionnaire (TIQ), and Hyperacusis Impact Questionnaire (HIQ): Factor Analysis, Test-Retest Reliability, and Minimum Detectable Change Using a Non-Clinical Population

PONE-D-24-56831R1

Dear Dr. Kula,

We’re pleased to inform you that your manuscript has been judged scientifically suitable for publication and will be formally accepted for publication once it meets all outstanding technical requirements.

Kind regards,

Prashanth Prabhu

Academic Editor

PLOS ONE

Additional Editor Comments (optional):

Reviewers' comments:

Reviewer's Responses to Questions

**Comments to the Author**

1. If the authors have adequately addressed your comments raised in a previous round of review and you feel that this manuscript is now acceptable for publication, you may indicate that here to bypass the “Comments to the Author” section, enter your conflict of interest statement in the “Confidential to Editor” section, and submit your "Accept" recommendation.

Reviewer #1: All comments have been addressed

Reviewer #2: All comments have been addressed

2. Is the manuscript technically sound, and do the data support the conclusions?

Reviewer #1: Yes

Reviewer #2: Yes

3. Has the statistical analysis been performed appropriately and rigorously? 

Reviewer #1: Yes

Reviewer #2: Yes

4. Have the authors made all data underlying the findings in their manuscript fully available?

Reviewer #1: Yes

Reviewer #2: Yes

5. Is the manuscript presented in an intelligible fashion and written in standard English?

Reviewer #1: Yes

Reviewer #2: Yes

6. Review Comments to the Author

Reviewer #1: The manuscript provides valuable information on the impact of tinnitus, hyperacusis, and misophonia in a non-clinical population. All reviewer comments have been adequately addressed by the authors.

Reviewer #2: The authors have carefully addressed all the reviewer's comments. I recommend acceptance of the manuscript.

7. PLOS authors have the option to publish the peer review history of their article (what does this mean? ). If published, this will include your full peer review and any attached files.

**Do you want your identity to be public for this peer review?** For information about this choice, including consent withdrawal, please see our Privacy Policy .

Reviewer #1: No

Reviewer #2: **Yes: ** Lipika Sarangi

---

## [Editor Report · Acceptance letter]

PONE-D-24-56831R1

PLOS ONE

Dear Dr. Kula,

I'm pleased to inform you that your manuscript has been deemed suitable for publication in PLOS ONE. Congratulations! Your manuscript is now being handed over to our production team.

Kind regards,

on behalf of

Dr. Prashanth Prabhu

Academic Editor

PLOS ONE